# Genomic Population Structure of the Main Historical Genetic Lines of Spanish Merino Sheep

**DOI:** 10.3390/ani12101327

**Published:** 2022-05-23

**Authors:** Antonio Granero, Gabriel Anaya, Sebastián Demyda-Peyrás, María J. Alcalde, Francisco Arrebola, Antonio Molina

**Affiliations:** 1Asociación Nacional de Criadores de Ganado Merino (ACME), 28007 Madrid, Spain; secretarioejecutivo@razamerina.com; 2Department of Genetics, University of Cordoba, CN IV, km 396, 14071 Cordoba, Spain; b22ancag@uco.es; 3Departamento de Producción Animal, Facultad de Ciencias Veterinarias, Universidad Nacional de La Plata, Calle 60, 118 s/n, La Plata 1900, Argentina; sdemyda@fcv.unlp.edu.ar; 4Consejo Nacional de Investigaciones Científicas y Técnicas (CONICET), La Plata 1900, Argentina; 5Agronomy Department, University of Sevilla, 41013 Sevilla, Spain; aldea@us.es; 6Agriculture, Livestock and Fisheries Research Institute (IFAPA), Hinojosa del Duque, El Viso Route, km 2, 14270 Cordoba, Spain; franciscoa.arrebola@juntadeandalucia.es

**Keywords:** sheep, merino, genetic lines, genomic characterization, SNPs

## Abstract

**Simple Summary:**

Historical documentation shows that the Spanish Merino sheep was selected over many centuries due to the quality of wool, following which it was used to originate all other Merino breeds around the world, mainly by crossbreeding with local breeds. Today, the historical genetic lines that originated the Spanish Merino are still preserved in several closed herds in which they have been bred for nearly 200 years, maintaining their original genetic purity. Our study demonstrates, using a genomic approach, the exceptional genetic richness and variability of these lines, which are clearly differentiated from modern Merino breeds, and must therefore be protected to safeguard the large genetic pool they represent.

**Abstract:**

According to historiographical documentation, the Romans first began to select Merino sheep in the Iberian Peninsula during the first century, with the aim of obtaining a breed appreciated for the quality of its wool. This process continued locally during the Middle Ages, when Spanish sheep were protected, and their export to foreign countries was banned. It was during the 16th century when individual Merino sheep were allowed to spread around the world to be used to improve the wool quality of local breeds. However, the wool crisis of the 1960s shifted the selection criteria of the Merino breed towards meat production at the expenses of wool. Consequently, individuals that display the genetic and phenotypic characteristics of those sheep originally bred in the kingdom of Spain in the Middle Ages are extremely difficult to find in commercial herds. In this study, we characterized the genetic basis of 403 individuals from the main historical Spanish Merino genetic lines (Granda, Hidalgo, Lopez-Montenegro, Maeso, Donoso and Egea), which were bred in isolation over the last 200 years, using a genomic approach based on genotyping data from the Axiom™ Ovine 50K SNP Genotyping Array. Our analysis included measuring population structure, genomic differentiation indexes, runs of homozygosity (ROH) patterns, and an analysis of molecular variance (AMOVA). The results showed large genetic differences between the historical lines, even though they belong to the same breed. In addition, ROH analysis showed differences due to increased inbreeding among the ancient generations compared with the modern Merino lines, confirming the breed’s ancestral and closed origin. However, our results also showed a high variability and richness within the Spanish historical Merino lines from a genetic viewpoint. This fact, together with their great ability to produce high-quality wool, suggests that ancestral Merino lines from Spain should be considered a valuable genetic population to be maintained as a resource for the improvement of wool-producing sheep breeds all around the world.

## 1. Introduction

The Spanish Merino is one of the most important sheep breeds in the world, particularly due to the high quality of its wool, but also due to its economic and historical importance, to the point where it was involved in the origin of most current wool-sheep breeds, worldwide. However, Merino individuals have also been characterized for their ability to thrive in highly diverse and harsh environments and weather conditions around the world, which is why they are frequently employed in combined meat–wool productive systems. In 2020, the Spanish Merino census included 121,727 breeding males and females located in 7457 different herds, although most of them (approximately 87% of individuals) were located in the dry, arid areas and pasturelands of the south and southeast of the country [1].

The Spanish Merino is one of the oldest and toughest sheep breeds in the world. Although the origin of the Merino is not entirely clear, there is Roman documentary evidence that indicates that the Iberian Black Merino is the ancestral origin of this breed [2,3,4]. According to this, these dark-coated animals were crossed by the Romans with foreign individuals obtained from the Atlas plateaus (North of Africa), which were highly appreciated for their coat, but also highly adapted for enduring extreme variations in weather conditions, including temperatures ranging from −20 °C to 35 °C [5]. These crossbred individuals were further selected by the Romans during the second century to produce white wool. This further selection marks the beginnings of the breed, as illustrated by the discovery of reliefs of a male Merino sheep’s head on a third-century Roman sarcophagus in an archeological excavation in the city of Cordoba, Spain [6]. Thereafter, these individuals were further selected for several centuries in Spain, focused on one particular trait—the fineness of the wool [7]—producing a reduction in the diameter of the wool fiber by approximately 25%, but also an increase in the weight of the fleece. For this reason, Merino sheep were considered at that time to be the first worldwide “industrial breed” [8,9,10].

To protect this valuable, sought-after breed, King Alfonso X founded the Honorable Council of the Mesta, which in the 13th century banned the export of ewes to other countries for several centuries, with the sole exception of Portugal in the 16th century [4,11]. Around five hundred years later (in the early 18th century), this prohibition was lifted, and Merino sheep began to spread the quality of their wool around the world. During this period, Merino rams were exported to several European countries to improve local sheep breeds, including Russia (1720), France (1725), Germany (1765), Austria (1775), Italy (1793), the Netherlands (1786), Denmark (1797), Sweden (1715), Hungary (1774) and Britain (1792) [8,9,12,13,14,15,16,17]. Later, in the 19th century, the Spanish Merino crossed the oceans, being exported to more distant countries, such as South Africa, Australia, the United States of America, and Argentina [10,18]. For these reasons, it is said that the Spanish Merino has played a major role in the origin of several breeds around the world.

The economic crisis that hit the wool industry in the 1960s affected the Spanish Merino breed to a greater extent, since its selection criteria shifted towards meat production in an attempt to obtain an alternative source of income other than wool [19]. To achieve this objective, foreign breeds with higher meat aptitude, such as the Mérino précoce, the Merine of Landschaf, the Merine of Fleischschaf and the Ille de France began to be bred in Spain, and were used by some breeders to cross with their flocks, which were not being bred in purity [20]. For this reason, several ancient Spanish genetic lines that had focused only on wool production disappeared from commercial herds [21]. However, some of them have survived thanks to efforts from several groups of traditional breeders who decided to continue selecting individuals for their wool quality in closed breeds, just as they had been bred for the previous 200 years. Despite the fact that these individuals are not officially acknowledged as strains, a simple morphological study makes it possible to determine the existence of significant morphological and productive differences [8,9,19,22]. Among them, the most important Merino lines selected for wool production that are still bred in Spain are the Maesso, Egea, Granda, Lopez-Montenegro, Hidalgo, and Donoso.

The Donoso line was formed from the historical flock of the Donoso family, which, since the 15th century, has managed a large flock of several thousand transhumant sheep. Six centuries later, the Donoso line was stabilized, and included individuals characterized by good quality meat production, while keeping most of the original wool quality. Currently, the Donoso sheep is characterized by being eumetric, with a very good conformation and meat aptitude, while also producing wool of medium fineness, exceptional fiber length and performance, and which is free of wool-grease.

The Egea line originated from a royal flock of 35,000 transhumant sheep from three different herds, which at the beginning of the 19th century was stabilized to 3000 sheep and 1000 rams. Egea sheep are characterized by being individuals of medium size that are sturdy and short-legged, with good meat production and maternal aptitude. Their wool is also characterized by a medium fineness and being free of grease, with a remarkable wick length covering the sheep’s limbs and heads.

The Granda line origin was associated with the Count of Oliva royal flock in 1845, although its origin might go back further. Currently only two pure-bred herds remain to the present day. Granda sheep are horned, eumetric and very well proportioned, producing a medium fineness but very uniform wool free of grease. Over the last few decades, there has been some degree of cross-breeding with Hidalgo rams.

The Hidalgo line origin was associated with the royal flock of the famous Duke of Fernán Núñez in the 19th century, which was one of the most important lineages in the first half of the 20th century, with around 3000 transhumant sheep, but nowadays, only two pure-bred herds remain. Hidalgo sheep are large with long limbs and closed hocks, without wool covering most of their face and legs. Hidalgo wool is characterized by its extreme fineness, but also by its short wick and grease content, which gives Hidalgo sheep a characteristic blackened appearance. Hidalgo sheep are also known for a marked maternal character, with increased production of milk for Merino sheep.

The López-Montenegro lineage has a family origin, being mentioned as differentiated livestock in some references from the mid-18th century [23]. This line originated from a transhumant flock of 4000 sheep, but since the middle of the last century, it has reached a stable size of 6000 individuals. Lopez-Montenegro sheep are characterized by being eumetric, with a slight tendency towards being ellipometric, covered by a fleece of long and fine fiber that is relatively free of wool-grease. This line is also known to have a natural tendency to form wool ties.

Finally, the Maesso line was originated within the family stock in the 16th century. In the 19th century, however, the line underwent exponential growth, reaching as many as 25,000 grazing animals. Maesso sheep are small, with a tendency towards being ellipometric, very rustic and well-adapted to harsh environments. They are also characterized by the fine quality of their wool.

All of these traditional Merino lines are considered to have been endangered in recent decades, due to a dangerous decrease in their population, mainly due to the shift in the breeding system from wool to meat production. For this reason, the Spanish Government decided 50 years ago (in 1971), to establish an ex-situ conservation program in Hinojosa del Duque (Cordoba, Andalusia), where individuals of each of these six ancient Merino lines (Maesso, Egea, López-Montenegro, Granda, Hidalgo and Donoso) are bred in purity to the present day.

Nowadays, high-throughput genomic methodologies, such as the availability of SNP chips, have made it possible for us to determine the population structure and variability from a genetic viewpoint with accuracy and reliability [24,25,26]. As an example, recent studies have been able to characterize different sheep breeds worldwide by using medium-density chips [13,15,27,28,29,30,31,32]. Some of these have even included Spanish Merino individuals from different locations in Spain (such as Andalucía and Extremadura) [13]. However, to our knowledge, there are no specific studies analyzing the ancient Spanish Merino lines involved in the beginnings of the breed.

The present study aims to evaluate the population structure, genetic variability, and differentiation situation of the last historical genetic lines of the Spanish Merino sheep, and to compare them with other Merino breeds from around the world using genomic data.

## 2. Materials and Methods

### 2.1. Animal Sampling

Samples from 403 Spanish Merino sheep belonging to the Asociación Nacional de Criadores de Ganado Merino (ACME) were analyzed in this study. The first dataset analyzed (dataset I) included individuals from six historical Spanish White Merino (ME) genetic lines: Granda (GRA, *n* = 60), Hidalgo (HID, *n* = 60), López-Montenegro (LM, *n* = 50), Maesso (MAE, *n* = 57), Donoso (DON, *n* = 60) and Egea (EG, *n* = 58), which have been bred in closed farms for up to 200 years (about 70 generations) according to the existing records. In addition, 20 Spanish White Merino individuals randomly selected from commercial herds (SMEI, *n* = 20) were included as an undifferentiated control, and 39 Spanish Black Merino (MN) were included as outgroup control within the breed. Individuals within groups and group sizes were selected from all of the available animals on the ACME database avoiding close kinships, following the Food and Agriculture Organization of the United Nations recommendations for the genomic characterization of animal genetic resources guidelines [33]. In all cases, biological samples (whole blood) were collected using EDTA-K3 BD vacutainers™ (BD, Madrid, Spain) by ACME technicians.

### 2.2. Genotyping and Quality Control

Genomic DNA was isolated from blood using the commercial DNA purification kit DNeasy Blood & Tissue Kit (Qiagen, Germantown, MD, USA) following the manufacturer’s instructions. Thereafter, the individuals were genotyped using the Axiom™ Ovine 50K SNP Genotyping Array (Thermofisher, Waltham, MA, USA), including 56,250 markers. Genotype calls were obtained by analyzing raw data files following the best genotyping practices workflow procedure in the Axiom analysis suite package v5.0 [34]. All of the samples showed high-quality genotyping results (DQC ≥ 0.82 and individual call rate QC ≥ 0.90) and therefore were included in this study.

### 2.3. Construction of a Worldwide Merino Breeds Database

To analyze the genetic differentiation of the Spanish Merino ancient lines with different Merino breeds in the rest of the world, we created a second dataset (dataset II) by merging our data with 194 genotypes obtained from public repositories [15]. The non-Spanish genotypes included Australian (AM, *n* = 100), French Rambouillet (RMB, *n* = 50), German Landschaf (MELAND, *n* = 21), and Chinese (CHIME, *n* = 23) Merinos as well as the genotypes of 24 Mouflons (*Ovis gmelini*, SMU) collected in a hunting state from the south of Spain as an outgroup species control.

Both datasets were first pruned by removing all of the markers showing a minor allele frequency (MAF) < 0.01 or without physical annotation. Thereafter, those SNPs in linkage disequilibrium (LD) with a variance inflation factor (VIF) of 1.25 were also removed. The final dataset included 28,765 SNPs. All data curation was performed using PLINK V2.0 program [35].

### 2.4. Population Structure and Genomic Differentiation

To compile a profile of the genetic population structures and their stratification, we first measured expected and observed heterozygosity using the dartR package [36] from the R statistical environment v4.1.3 [37] as well as a Principal Component Analysis using PLINK V2.0 [35]. In a further analysis, we estimated the probability of adscription of each individual to their subpopulations using a Bayesian unsupervised analysis implemented in *Admixture* software v1.3 [38]. The optimal K value was determined as the one having the lowest cross-validation (CV) error. Finally, a phylogenetic analysis based on the genetic distance of Hamming at individual and subpopulation levels was performed using the *ape 5.5* package [39] in R. The results were visualized in radial and non-radial trees.

### 2.5. Analysis of ROH Patterns

The existence of genetic fingerprints within the Spanish White Merino was determined by estimating the existence of genomic regions in which the run of homozygosity (ROH) abundance had significantly increased. To this, we first estimated ROH per individual using a 7-class model in the RZooROH R package [40]. Krate limits were set at 6, 12, 18, 36, 144, and 144. The mixing coefficient was set at 0.01, and genotyping error probability at 0.001. After processing, only ROH longer than 1Mb were retained for further analysis to avoid the detection of false positives due to LD [41]. In a second step, the quantitative analysis of the ROH fragments was performed using the DectectRuns package in R [42], which determined an ROH-based inbreeding value (F_ROH_) per individual at the chromosome level, as the percentage of the bases of each respective chromosome covered by ROH [43]. In addition, we estimated the F_ROH_ values theoretically produced by a mating that occurred 3, 6, and 9 generations ago (F_ROH3G_, F_ROH6G_, and F_ROH9G_) by analyzing ROH longer than 16 Mb, 8.8 Mb, and 5.5 Mb only, according to Fisher’s theory [44]. In addition, an ancestral value (F_ROHANC_) was also estimated by analyzing ROH shorter than 5.5 Mb. Finally, an incidence p-value at the SNP level was determined using a 1,000,000-iteration permutation test following the methodology described by Goszczynski, et al. [45]

### 2.6. Analysis of the Molecular Variance (AMOVA)

To estimate the percentage of the variance components explained by the species, breeds, and genetic lines using genomic data, we performed an analysis of molecular variance (AMOVA, [46]). The bioinformatics workflow, developed in R, included an initial transformation of the binary genomic dataset (bed, bim, and fam) into a *genind* object using the radiator package [47]. Thereafter, we performed an AMOVA (*poppr.amova*) implemented in the Poppr R package [48]. The model (STRATA) included 3 levels of differentiation: species, breed, and line.

## 3. Results and Discussion

The development of high-throughput genotyping technologies, in which allelic variants of thousands of SNP markers can be easily determined at a very low cost, has opened a new era in the genetic characterization of animal populations [49]. In recent years, these methodologies have been widely employed in sheep [50,51], including several studies in which the Spanish Merino was included as a part of inter-breed analysis [13,15,29,31,52,53]. However, no specific studies have yet been performed assessing the genetic variability of the ancient Merino Spanish lines, which are considered one of the closest populations to the original Merino breed remaining in the present day.

The initial estimation of genomic variability based on the observed (Ho) and expected (He) among the populations and lines are presented in Table 1. The analysis of the whole dataset showed Ho = 0.363 and He = 0.374, which ranged at the populational level between 0.341 (RMB) and 0.363 (ME) in Ho, and between 0.349 (MN) and 0.375 (AM) in He. In AM, CHIME, RMB, and ME, the observed heterozygosity was similar to what was expected, suggesting that little repercussion of the genetic selection applied to the loss of variability. On the contrary, MELAND and MN observed heterozygosity was higher than expected, suggesting the existence of measures to stop the increase in F due to its low effective size. Interestingly, the results obtained in the Spanish wild mouflon, which arrived in the south of Spain in the middle of the 20th century, derived from individuals introduced to Italy around 7000 years ago as a wild population [54,55,56], were much lower (Ho and He were 0.334 and 0.312, respectively).

Similar results were observed in the analysis of the ancient and modern lines within the Spanish Merino, in which the commercial Merino SMEI had the lowest value of Ho (0.331), while the LM had the highest (0.375). In DON, EG, MAE and LM, the Ho was higher than the He, while GRA, HID and SMEI showed the opposite. In a previous study, Azor, Cervantes, Valera, Arranz, Medina, Gutiérrez, Goyache, Muñoz and Molina [21] estimated Ho and He values in four of the historical lines (GRA, HID, LM and DON) using genomic information. In both cases, HID and LM showed the lowest and highest Ho values, respectively, among the ancestral lines, but in the former, the He/Ho relation was lower than 1 in the four lines. Fifteen years later, the same genetic trend remains only in the GRA and HID, suggesting that the heterozygosity and genetic variability were preserved adequately within those ancient lines, probably as a consequence of a rationale crossing management among the breeders in the different herds. However, this trend was not observed in the rest of the genetic lines, in which a decrease in variability was observed.

However, it is worth mentioning that Azor, Cervantes, Valera, Arranz, Medina, Gutiérrez, Goyache, Muñoz and Molina [21] employed genomic data obtained from 33 STR markers recommended by the FAO [57] for the estimation of genetic variability in sheep populations (no SNP arrays were available at that time). Therefore, the comparison with our results, obtained using close to 56,000 SNP markers, needs to be taken with caution. However, Fernández, et al. [58] demonstrated that three SNPs per STR are needed to obtain an equivalent Q value and the same matching probability in a parentage test, but also that the heterozygosity levels were higher in the same individual when they were genotyped with microsatellite instead of SNPs. This is also in agreement with the lower heterozygosity observed in the present study in the same population in comparison with the situation reported 15 years ago by Azor, Cervantes, Valera, Arranz, Medina, Gutiérrez, Goyache, Muñoz and Molina [21]. In addition, the observed and expected heterozygosity values obtained in our study are located within the same range shown for other sheep breeds in several studies performed employing 50K SNP chips [15,59,60,61], supporting the reliability of our conclusions.

Genomic analysis based on SNP data makes it possible to estimate differences in populations based on inbreeding patterns very accurately, by analyzing the incidence at the genomic level of Runs of Homozygosity (ROH) [62]. Figure 1A shows the Frequency of Runs of Homozygosity (F_ROH_) values in the analyzed population, averaging 0.037 in the six historical lines studied. Despite that the results can be considered to be low on average, there were large differences among the lines, with LM being the one with the lowest value (F_ROH_ = 0.017) and HID the one with the highest (F_ROH_ = 0.057). To differentiate ancient and recent inbreeding, we calculated F_ROH_ considering different ROH lengths grouped into four categories. LM was still the line with the lowest levels of homozygosity across all of the generations calculated, including the ancestral one (F_ROH_ Ancient = 0.0064), while HID produced the opposite values, except in the F_ROH_ calculated three generations ago, where the highest value appeared in the MAE population (F_ROH 3G_ = 0.0107). When analyzing the proportion of ROH in each line, the populations that accumulated the maximal percentage of ancient inbreeding were the HID and DON lines (42.54% and 41.18%, respectively), whereas the MAE line showed the lowest (25.53%) (Figure 1B). In addition, the MAE showed the maximum percentage of recent inbreeding and HID the lowest percentage (ROH 3G proportions of 36.47% and 13.01%, respectively). The evolution of inbreeding over generations three to nine followed the same trend in all of the populations. Except for LM and MAE, all of the other genetic lines showed strong inbreeding in the first generations analyzed, but after nine generations, the F_ROH_ decreased significantly due to management aimed at decreasing inbreeding (Figure 1A). On the other hand, MAE and LM showed a low initial F_ROH_ that increased greatly during some decades due to the low population size, although in the last decade, it has been controlled.

The degree of genetic differentiation among the populations was assessed through an analysis of molecular variance (AMOVA), which quantified the percentage of the differences explained by the genetic lines and breeds (Table 2). Differences among breeds (AM, ME, MELAND, CHIME, MN, and RMB) explained 5.01% of the genomic variance, while the differences among individuals within breeds accounted for 2.75%. Overall, the results suggest close genetic proximity between all of the breeds derived from the Merino trunk and a lower differentiation between the individuals from each of them.

Interestingly, the Spanish Merinos showed a lower variability among lines (3.99%), but with almost no variation explained by individuals (0.12%) within the lines. Even though the differentiation between lines can be considered high (even close to that observed among breeds), the lack of variability within them supports the existence of six “pure” ancient genotypes that are clearly differentiated. In addition, the differences between individuals of the Merino lineages are also much smaller than the differences between individuals of the Merino-derived breeds, which is in agreement with the closed breeding practices developed during the formation of the historical Merino lines.

Principal Component Analysis (PCA) of dataset II revealed a clear clustering of the animals according to their breed (RMB, AM, MELAND, and MN), in most cases (Figure 2A). However, the CHIME cluster showed, apart from the main group of individuals, a second, smaller one located near the RMB. In addition, Spanish Merino animals (ME) showed a more dispersed clustering, which fits in with the wide variability and genetic richness observed among the different lines included within it in this analysis. In addition, it is worth mentioning that the first two PC components captured 42.63% of the total genomic variability (Figure 2B) (20.42% when the Mouflon outbred group was removed, Figure 2A). In both cases, the percentage of genomic variation captured by the first two PCA components is large, and therefore, the conclusion obtained from its analysis can be considered to be highly reliable.

The genetic differences between the wild Mouflon and domestic sheep breeds are observed, and some individuals of Spanish Merino and the RMB sheep are the two most distant genetic groups. On the other hand, the MN was the nearest breed to the Mouflon, which is logical, since it is the most ancestral Merino, a fact which could place these animals as the inception population of the breed [5,8].

Another interesting finding was the existence of a close genetic distance between Black Merinos and Mouflon, since it is well known that MN was probably the most significant ancestral breed in the Merino trunk [2]. However, MN is also located between Spanish Merino and non-Spanish Merino, with the sole exception being a group of DON individuals, located near the Australian Merino. Both groups of individuals have been highly selected for wool production.

In the same way, the Spanish White Merino was located closer to the German Merino than to the Australian Merino. Two hypotheses can explain this finding. First, the introgression of the Spanish Merino into the German Merino occurred in 1765, while the same situation occurred in the Australian Merino over 100 years later (late 19th century). Furthermore, the genetic admixture between Spanish and German Merino breeds has continued on a large scale, with several events documented only 60 years ago (in the 1950s and 1960s).

Our data revealed a high degree of heterogeneity in the Chinese Merino, which fits with the origin of the breed by crossing Australian Merinos with various Chinese sheep breeds. For example, the Chinese Merino partly originated from an absorbent crossing with Gansu Mountain Sheep, generating a new breed with fine wool and high rusticity [63]. This would also justify the fact that it is an older population, closer to the Black Merino than other Merino populations worldwide; however, the results could also be explained by the lower intensity of selection to which CHIME was subjected.

The intra-breed analysis, including all of the Spanish Merino lines, as well as the Black Merino without Mouflon and with Mouflon, is shown in Figure 3A,B, respectively. The Mouflon was clearly differentiated from the rest of the breeds and lines (Figure 3B). In the refined analysis including only the ancient lines (Figure 3A), there is also a clear genetic differentiation among DON, EG, MAE, LM, and HID, although some overlaps are evident in a few individuals. However, it is worth mentioning that the first two principal components accounted for 18% of the genomic variance, which shows a clear differentiation among the lines. Interestingly, the animals from GRA formed two different clouds separated by the animals from the LM line. This could be explained by the accumulation of genetic differences between herds within the same line across generations produced by differences in herd and selection practices. In addition, MAE and EG lines are close but are highly separated from the HID line, and SMEI individuals are located in the middle of the six historical lines, which fits in with the high degree of admixture with the Spanish Merino breed. Finally, the Spanish Black Merino is separated from the rest of the ancient lines but shows some individuals located near the DON and the LM clusters, suggesting a close genetic relationship.

Admixture analysis also revealed clear differences in the genetic composition among breeds and lines. The first analysis (Figure 4, K = 2) showed a large genetic similarity between the Spanish historical lines, Black Merino and the Mouflon, compared to the rest of the Non-Spanish Merino, with the sole exception of the SMEI population, highlighting the ancient origin of the Spanish Merino lines. Conversely, the non-Spanish Merino breeds share less genetic homology with the Mouflon, since they are breeds selected much more recently. However, both results also support the hypothesis of the ancestral origin of the Black Merino and its involvement in the first derivation of the Spanish Merino in the origin of the entire non-Spanish Merino and Merino-derived breeds. This is also supported by the K = 3 analysis, in which most of the Spanish historical lines (EG, GRA, HID, LM, MAE) are close to the MN lineage, and SMEI and MN are more differentiated from the rest of the Spanish Merino lines. This aggrupation of the historical lines of the Spanish Merino and the Black Merino as a recognizable population differentiated from the other non-Spanish breeds is repeated in K 4 to K 6. In addition, K = 6 analysis depicts the degree of influence of the Spanish White Merino in the AM, CHIME, and MELAND, supporting the hypothesis of the Iberian origin of the Merino trunk, as was previously reported by Ciani, Lasagna, D’Andrea, Alloggio, Marroni, Ceccobelli, Delgado Bermejo, Sarti, Kijas, Lenstra, Pilla and the International Sheep Genomics [13]. Finally, in K = 7, the DON and HID lines are considered independent populations at the same level as MN and RMB. The animals of the different Merino breeds and the Mouflon we analyzed are not often structured into 12 populations according to the optimal K (CV error: 0.593). Remarkably, AM, CHIME, and SMEI have a high mix of different populations in their genetic compositions that are usually detected in industrial breeds in which admixture and crossbreeding are common. On the other hand, MELAND and RMB are highly homogeneous, probably due to the intense genetic selection that these populations underwent.

A phylogenetic distance tree including all of the Spanish Merino individuals analyzed in this study is shown in Figure 5. DON, MAE and EG individuals are clustered into their populations. GRA and HID individuals are aggregated in groups with different subpopulations, and animals from the SMEI and LM show greater dispersion. The substructure of the population shows how the SMEI animals appear mixed with the HID, LM and some GRA herds; this could be due to the SMEI group containing animals crossed with other breeds, in some cases, derived from the Merino [19]. Subsequently, SMEI breeders, have been able to use animals from pure herds to “re-merinize” their herds. The HID and GRA herds have been the most recurrent resources used to this end since both lines are highly popular ancient lines associated with the Merino characteristics among Spanish breeders.

Interestingly, some LM and GRA individuals are quite similar from a genetic point of view. This could be explained by the fact that the populations are bred in the same geographical area, and under similar management. For this reason, it can be assumed that these lineages could have formed centuries ago from the same old historic royal flocks, although the occurrence of a modern crossbreeding event cannot be ruled out either.

In contrast, GRA individuals showed two clear genetically divergent groups (which was also observed in the K = 12 admixture analysis). We speculate that these differences can be explained by the fact that GRA herds were bred in isolation in very distant geographical areas over the last 50 years, thus producing genetic differences by selective breeding despite their common origin.

To find the time of divergence (Figure 6), we measured the phylogenetic relations between the different breeds (Figure 6A)/lines (Figure 6B). The root shows how SMU, as the outgroup control in a single branch, is differentiated from the rest of the breeds. In the next node, MN appears in a single external branch followed by MELAND, RMB and CHIME. The next node shows the AM and the ME breeds on the same level. When ME is disaggregated according to its genetic line subpopulations (Figure 6B), the SMEI is located between the RMB and the CHIME. Behind the CHIME node stands the DON historical line of the Spanish Merino, and then, the next clade divides the AM with the rest of the Spanish Merino historical lines, MAE, EG, HID, respectively, and finally with the GRA and LM on the same level.

## 4. Conclusions

This is the first study to assess the genetic structure of the Spanish Merino breed and its most ancient and original genetic populations using a high-throughput genomic approach. Overall, we determined the existence of a high degree of genetic variability within the breed, as well as its influence in improving Merino breeds around the world. In addition, we demonstrated the importance of the ancestral Merino lines, which are still bred in purity in Spain, as a genetic reservoir of variability for improving the breed, as well as for conservational purposes. In addition, the existence of such rich genetic variability could provide the starting point for an association, in further studies, between the genomic variations and the phenotypes of each of these populations. This genetic richness, together with the excellent, highly sought-after aptitude of the wool, makes it essential to undertake all possible efforts aimed at preserving the different historical lines as a genetic source for the improvement of other derived Merino breeds. However, the conservation of these ancient genetic lines is also justified for historical and cultural reasons and for the different aptitudes that each of them presents, such as differences in grazing capacity, resistance to thermal stress, growth rate, meat quality, and maternal aptitudes, among others, which could be of interest soon. Therefore, it would be convenient to promote actions such as the incorporation of breeders from other herds and the creation of a germplasm bank of the best animals of each line in the Hinojosa del Duque Center (ex-situ conservation), where most of these pure lines are still bred in purity.

## Figures and Tables

**Figure 1 animals-12-01327-f001:**
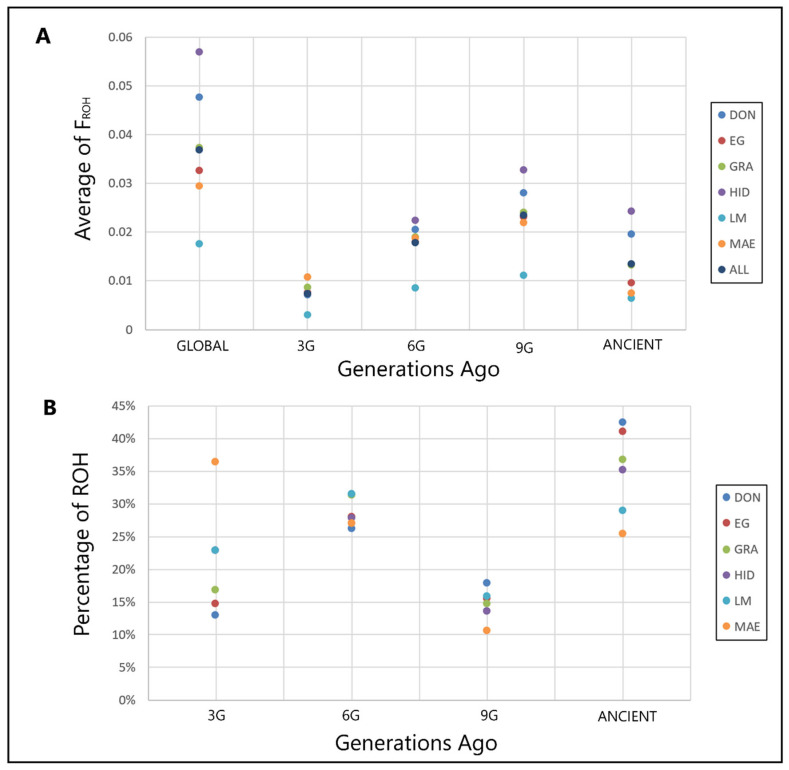
(**A**) Average of F_ROH_ of the historical genetic lines of the Spanish Merino. F_ROH_: genomic inbreeding coefficients; GLOBAL: global F_ROH_; 3G: F_ROH_ of 3 generations ago; 6G: F_ROH_ of 6 generations ago; 9G: F_ROH_ of 9 generations ago; ANCIENT: F_ROH_ of over 9 generations ago. (**B**) Percentage of Run of Homozygosity (ROH) by class. ROH: Run of Homozygosity; 3G: Percentage of ROH 3 generations ago; 6G: Percentage of ROH 6 generations ago; 9G: Percentage of ROH 9 generations ago; ANCIENT: Percentage of ROH over 9 generations ago. Historical lines: DON (Donoso), EG (Egea), GRA (Granda), HID (Hidalgo), MAE (Maesso), LM (López-Montenegro), ALL (all of the genetic lines together).

**Figure 2 animals-12-01327-f002:**
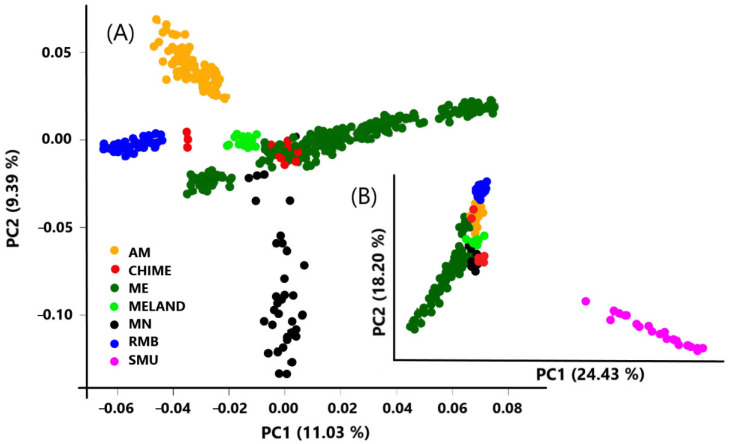
Genomic principal component analysis of the Merino-derived population with Mouflon (**B**) and without Mouflon (**A**) as outgroup. Breeds: AM (Australian Merino), CHIME (Chinese Merino), MELAND (Landschaft Merino), RMB (Merino of Rambouillet), MN (Spanish Black Merino), and ME (Spanish White Merino). Outgroup species: SMU (Mouflon).

**Figure 3 animals-12-01327-f003:**
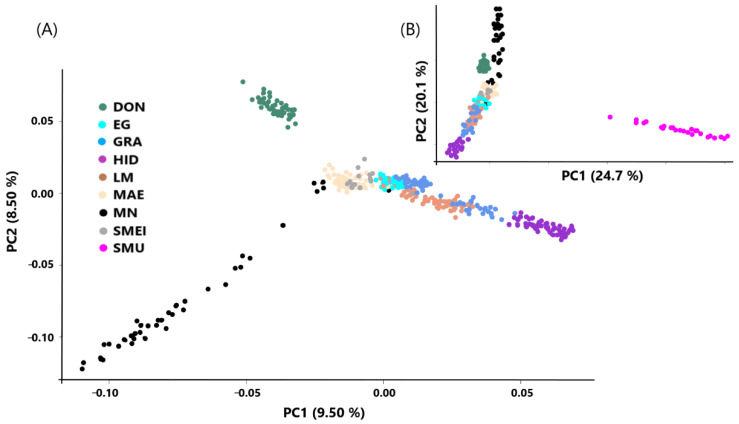
Principal component analysis of the Spanish Merino lines and the Black Merino breeds population using the Mouflon (**A**) and without (**B**) as outgroup. (**A**) Genetic lines within the Spanish Merino breed: DON (Donoso), EG (Egea), GRA (Granda), HID (Hidalgo), MAE (Maesso), LM (López-Montenegro), SMEI (Spanish Merino from Industrial Breeding). Outgroup species: SMU (Mouflon). (**B**) Detailed PCA of the Spanish Merino genetic lines and the Black Merino without the Mouflon.

**Figure 4 animals-12-01327-f004:**
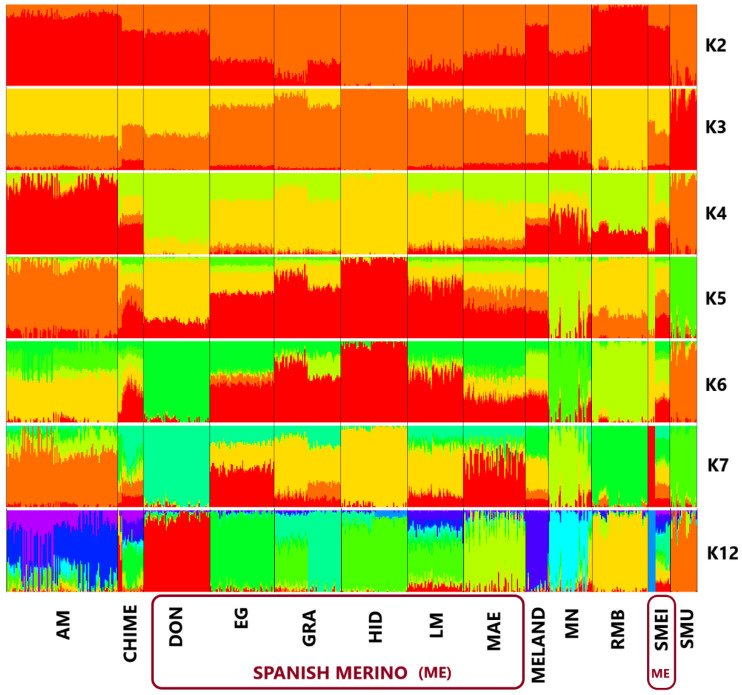
Plot of the population structure of the Merino breeds including the genetic lines of the Spanish Merino and the Mouflon employed in the study according to different K, including the optimal K (K = 12). K: number of populations assigned. Breeds: AM (Australian Merino), CHIME (Chinesse Merino), MELAND (Landschaf Merino), RMB (Merino of Rambouillet), MN (Spanish Black Merino). Genetic lines within the Spanish Merino: DON (Donoso), EG (Egea), GRA (Granda), HID (Hidalgo), MAE (Maesso), LM (López-Montenegro), SMEI (Spanish Merino from Industrial Breeding). Outgroup species: SMU (Mouflon).

**Figure 5 animals-12-01327-f005:**
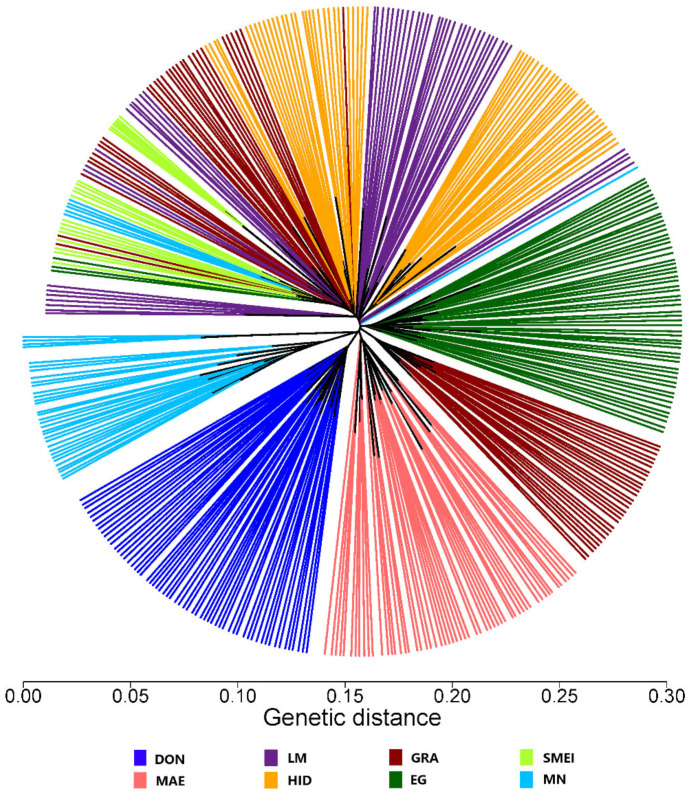
Unrooted tree of genetic distances between individuals of the historical lines of the Spanish Merino and MN. Genetic lines within the Spanish Merino: DON (Donoso), EG (Egea), GRA (Granda), HID (Hidalgo), MAE (Maesso), LM (López-Montenegro), SMEI (Spanish Merino from Industrial Breeding). Breed: MN (Spanish Black Merino).

**Figure 6 animals-12-01327-f006:**
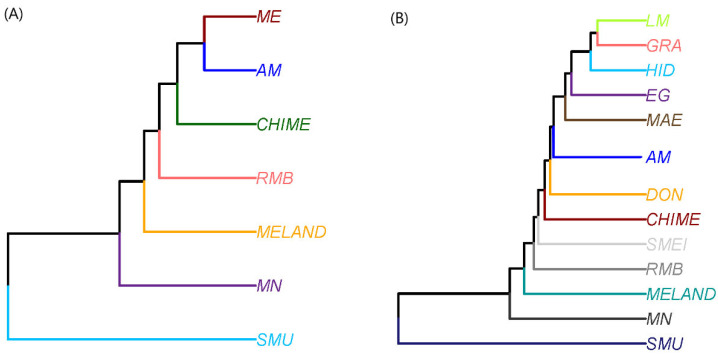
(**A**) Pop tree of the different breeds and mouflon as the outgroup control using genomic phylogenetic analysis. (**B**) Pop tree of the different breeds and mouflon as the outgroup control using genomic phylogenetic analysis. Tree generated with the genetic distances of NEI using the UPGMA method. Breeds: AM (Australian Merino), CHIME (Chinese Merino), ME Spanish Merino, MELAND (Landschaft Merino), RMB (Merino of Rambouillet), MN (Spanish Black Merino). Genetic lines within the Spanish Merino: DON (Donoso), EG (Egea), GRA (Granda), HID (Hidalgo), MAE (Maesso), LM (López-Montenegro), SMEI (Spanish Merino from Industrial Breeding). Outgroup species: SMU (Mouflon).

**Table 1 animals-12-01327-t001:** Observed (Ho) and expected heterozygosity (He) of the analyzed population (breed, genetic lines, and species). Breeds: AM (Australian Merino), CHIME (Chinese Merino), MELAND (Landschaft Merino), RMB (Merino of Rambouillet), MN (Spanish Black Merino), and ME (Spanish White Merino). Genetic lines within the ME breed: DON (Donoso), EG (Egea), GRA (Granda), HID (Hidalgo), MAE (Maesso), LM (López-Montenegro). Undifferentiated Spanish Merino SMEI (Spanish Merino from Industrial Breeding). Outgroup species: SMU (Mouflon). N, number of individuals. Ho, observed heterozygosity. He, expected Heterozygosity.

Population	Breed	N	Ho	He
AM	Australian Merino	100	0.361	0.375
CHIME	Chinese Merino	23	0.358	0.367
MELAND	Landschaf Merino	20	0.358	0.356
RMB	Rambouillet Merino	50	0.341	0.356
ME	Whole White Spanish Merino	364	0.363	0.374
DON	Spanish Merino	60	0.359	0.357
EG	Spanish Merino	58	0.367	0.364
GRA	Spanish Merino	60	0.366	0.367
HID	Spanish Merino	60	0.355	0.357
MAE	Spanish Merino	56	0.369	0.366
LM	Spanish Merino	50	0.375	0.372
SMEI	Spanish Merino	20	0.331	0.358
MN	Spanish Black Merino	39	0.358	0.349
SMU	Mouflon	24	0.334	0.312

**Table 2 animals-12-01327-t002:** Analysis of molecular variance (AMOVA) of the Merino-derived population using genomic data. A (% of Variation between Pop), B (% of Variation between samples within Pop), C (% of Variation within samples).

	N	N Pop	% of Variation		
			A	B	C
Merino derived Breeds ^1^	620	6	5.01	2.75	92.23
Spanish Merino genetic lines ^2^	364	7	3.99	0.12	95.89

^1^ Breeds: AM, ME, MELAND, CHIME, MN and RMB. ^2^ Spanish Merino historical genetic lines (DON, EG, GRA, HID, MAE, LM) and Spanish Merino with commercial purposes (SMEI).

## Data Availability

The data sets employed in this study are the property of the Asociación Nacional de Criadores de Ganado Merino (ACME) and were provided for scientific pur-poses under a specific collaboration arrangement. The data set could be provided for scientific purposes to further authors under reasonable request in the ACME technical department.

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
