# Peer review of "Genomic Population Structure of the Main Historical Genetic Lines of Spanish Merino Sheep"

_animals, 2022, doi:10.3390/ani12101327_

Round 1

Reviewer 1 Report

Although the results presented in your manuscript are interesting, because it is the first one made using SNPchips. However, your manuscript presents mainly descriptive research on levels of genetic diversity of a particular species.In particular, your manuscript would benefit considerably if it is framed to tests specific hypotheses, e.g.,  and provide some insight about the biological causes of those differences, as you indicated that they show distinct levels of genetic diversity. The work would improve a lot if the following observations are clarified and corrected.

Simple Summary: These farms have been closed for 200 years, which means that there is no genetic exchange with other farms, and the animals are sold or slaughtered, which means a loss of genetic diversity. So how do you justify that there is genetic diversity? In addition, other Ovis gmelini species cannot be included in the genetic structure analysis, because they have a different genome. 
Ovis gmelini, written in italics.

Line 36. change study for analysis
Line. 40. Change "enormous" to high.
Line. 124. What do they mean by "differentiated cabin"
Line 126. Change the citation format (Rodríguez, 2001).
Line 138-141. Please rewrite the paragraph: the Hinojosa del Duque herd was integrated with individuals from the Maesso, Egea, López-Monte-140 Negro, Granda, Hidalgo and Donoso lines?
Line 163-164: Why did you use undifferentiated control and outgroup control? Explain it.
Line 181. Why did they integrate merino breeds from other parts of the world? If what they want is to evaluate the populations of Spain?... First of all, there must be a genetic differentiation, because these breeds are crossed with other local breeds.
Line 240: 0.363 and 0.38, change (0.363 to 0.384 respectively), because there are more than two populations.
Line. 238. Comparisons of He should be made with other studies using the same markers, and not with microsatellite-type markers. Due to the nature of the markers, they may differ in levels of genetic diversity.
Line 260: table 1. It can be seen that the values ​​of Ho and He are in the same range in most of the populations, with small differences with respect to the control populations. How do you explain it? Does this really represent high levels of genetic diversity? How do you know that there is a loss of genetic diversity? Perhaps they should have included an indigenous breed as a control.

To know the levels of homozygotes and heterozygotes,In statistical analyses, a Hardy-Weinberg equilibrium (HWE) table should be included.

In the discussion, you should reference other studies done with other breeds using 50K SNP Genotyping. In this way they can have an idea of the high or low levels of genetic diversity, because it corresponds to the same species, limited to the merino breed as a reference of levels of genetic diversity. In other breeds of ovis aries, what are the levels of high genetic diversity that have been reported?

In the Conclusions, what conservation strategies do you recommend to maintain the levels of genetic diversity in the population of Spanish Merino?

In References, they should include DOI literature, and avoid manuals such as the FAO.

Reviewer 2 Report

Dear authors,

the manuscript reads pretty well. It is well written, the aim of is interest, and the analyses and results seem coherent.

I have just one suggestion since in the Introduction you mentioned that Merino animals are raised in different environments. Please consider modelling the differences not only among lines but also among environments (or at least to comment on that in the discussion). Some recent papers (https://doi.org/10.1186/s12711-021-00616-3; https://doi.org/10.3389/fgene.2021.604823; https://doi.org/10.1111/jbg.12666; https://doi.org/10.1007/s10592-018-1099-y) investigated the differences in selection signatures in animals raised in different places. It would be interesting to find traces of environmental adaptation, for example comparing lines raised in mountainous areas vs lines raised in the plains. These differences are more important to explain differences among lines/groups within a breed.

Line 48: “significant” should be “important”?

Lines 49-50: please check the sentence after “:”

Line 259, Table 1: please change the comma in “0,0118” with a full stop

Lines 265-266: Why P>0.001 should be “no significant differences”? The only error probability you mentioned was in “ROH patterns”

Lines 278-279: please explain the “6.94 to 7.97” alleles per markers. You cannot have 7 different alleles per markers. You can have different combinations of alleles, or different alleles or variants, within a gene (or haplotype or whatever) but the maximum number of alleles per SNP is 2 in livestock (one from the sire and one from the dam). You must explain this number.

Line 479, Figure 5: I personally do not like too much these kind of figures because they are not too clear, but I know that they are often used.

Best regards
